# Men’s Behavior and Communication in the Days Prior to a Suicide—A Psychological Autopsy Study

**DOI:** 10.3390/ijerph20176668

**Published:** 2023-08-28

**Authors:** Laura Hofmann, Birgit Wagner

**Affiliations:** Department of Clinical Psychology, Medical School Berlin, 14197 Berlin, Germany; birgit.wagner@medicalschool-berlin.de

**Keywords:** gender, masculinity, suicide prevention, mental health, psychological autopsy

## Abstract

Men show a significantly higher suicide rate, are less often recognized as persons at risk, and are more difficult to reach for suicide prevention interventions. Warning signs and deterioration in mental health are often not recognized by their surroundings. This study aims to retrospectively analyze the behavior and communication of men before a suicide, how relatives noticed changes, and how the interaction was perceived. *N* = 15 individuals who lost a close male relative to suicide were interviewed using psychological autopsy interviews. The interviews were evaluated following a deductive–inductive approach while using a comprehensive category system. The majority of men showed changes in behavior before the suicide, especially social withdrawal, irritability, and generally a deterioration in mental health. In fact, men did communicate their suicidal thoughts before they died through suicide, but mainly indirectly. While only one-third of the deceased made preparations before suicide, the majority of relatives noticed a deterioration in the mental health of the individual as well as increased alcohol and substance use. Men show signs of suicide, which are little recognized by their surroundings. Suicide prevention interventions should be adapted more to the needs of men.

## 1. Introduction

Suicide is a global phenomenon, claiming more than 700,000 lives a year [1]. In almost every country, suicides by men outnumber those by women. Globally, the suicide rate for men is estimated to be 1.8 times higher than that of women, resulting in men being three to four times more likely to die by suicide [1,2,3]. Although a gender paradox regarding suicide is evident, there is still too little research available on this topic, as well as a lack of effective interventions for suicide prevention in men. While it has been found that factors like a poor social network, the reluctance to seek help, stigma, as well as male stereotypes are associated with suicidal behavior in men [2,4,5], it is essential to thoroughly understand the behavior and specific warning signs of men prior to suicide to improve prevention strategies.

A large body of research has identified well-known symptoms of suicidal ideation and behavior, which primarily include hopelessness, sadness, social withdrawal, and mood swings [6,7]. However, one of the major problems in recognizing men at risk for suicide is the presence of atypical symptoms that this population group experiences in the context of suicidal ideation or mental health disorders, specifically regarding male depression. These atypical symptoms result in suicidality not being adequately identified, neither by their relatives nor by professionals. Many men express their distress prior to suicide through externalizing symptoms such as aggressiveness, antisocial and risk-taking behavior, irritability, and increased addictive behavior [8,9,10]. In their review, Hunt et al. [6] identified several factors and behaviors that men displayed prior to suicide. They found that men were often in a more positive mood in the days before the suicide compared to the weeks and months before. In most cases, men made hints about their suicidal thoughts beforehand, as well as showing a lack of problem-solving skills and increased anger in the time before the suicide.

Generally, the so-called gender paradox is spoken of regarding suicidal ideation and behavior in men and women [11]. According to this paradox, women more often attempt suicide, but men more often die by suicide. This is mainly due to the fact that men are less likely to share their distress and, thus, often do not receive adequate support from their social or professional environment. In addition, men have a significantly lower willingness to make use of support services than women, although the number of men who seek professional support in crises has risen continuously in recent years [12]. In addition, men use alcohol or other drugs for self-medication significantly more often [2,13]. Acute intoxication promotes disinhibition effects and reduces impulse control, which in turn promotes suicidal behavior.

Displaying atypical symptoms in men also results in an underdiagnosis of mental health disorders, as classic diagnostic systems often do not depict male-specific symptoms [14,15,16]. Depression, for example, is diagnosed two to three times more frequently in women than in men [17,18]. The so-called ‘male depression’ is recognized less frequently due to its own symptom pattern, which is not reflected in most diagnostic assessment instruments. Regarding relatives and friends, a prevailing image of how a person at risk for suicide behaves exists. As a result, even the individual’s close social circle often fails to grasp the severity of the symptomatology and misinterprets changes. However, it has also been shown that men themselves often do not recognize the severity of their own distress [19]. In addition to the already reduced willingness to seek help due to the prevailing masculinity norms, stigmatization, or fear of being seen as weak [12,20], men are less likely to feel they need psychosocial support, even if they report problems with their mental health [21].

Since men often have a smaller social support system and utilize much less the mental health care system, there are significantly fewer opportunities for them to communicate suicidal thoughts compared to women. While about 60–80% of all suicides are announced beforehand directly or indirectly, sometimes repeatedly, there seems to be no gender difference in the frequency of suicide-related communication [19]. Men, however, seem to prefer indirect communication, such as “I have had enough of everything”, resulting in suicidal ideation rarely being recognized from the outside [6,22]. However, it appears that men more often communicate their suicidal ideation with friends or with their partner, whereas women tend to contact the healthcare system, therefore becoming more likely to be identified as at risk [19]. Consequently, these gender-specific communication patterns prior to suicide mean that family members are often poorly informed about their relative’s suicidal ideation. They are, therefore, unable to identify any changes in male behavior or perceive communication as alarming [23]. These findings show that it is not only important to understand men’s behavior and symptom pattern but also to educate their surroundings regarding specific forms and communication of suicidal ideation and behavior in men.

To develop much-needed suicide prevention strategies, it is necessary to improve our understanding of men’s behavior in the days leading up to a suicide attempt. Through this, men at risk could be eventually better identified, and potential gatekeepers could be informed and receive psycho-education about these specific aspects of male suicidality. Gatekeepers are people who are potentially in contact with people at risk and can thus identify them and refer them to the support system. Gatekeepers are, for example, relatives, general practitioners, teachers, or work colleagues [24]. This study aims to identify how a sample of men behaved in the days leading to the suicide, using psychological autopsy interviews of close relatives who have lost a man to suicide. Specifically, we aimed to address the following research questions:(a)How did men behave in the prior days before a suicide?(b)Did men communicate their suicidal thoughts before a suicide?(c)Did men make any arrangements (e.g., finances, testament) before a suicide?(d)How did mental illness manifest itself in the men involved, and did symptoms change in the days before a suicide?(e)How did relatives perceive the men’s behavior?

## 2. Materials and Methods

### 2.1. Study Design

This study is part of the larger collaborative project *MEN-ACCESS: Suicide prevention for Men* consisting of the University of Leipzig, the Medical School Berlin, and the Bielefeld University. The research network aims to identify gender-specific risk factors, communication strategies, and gatekeepers. The present study follows a mixed-methods approach. Qualitative data were collected by conducting psychological autopsy interviews with bereaved individuals who lost a male relative or partner to suicide. Quantitative data were obtained prior to the interviews using online questionnaires. Psychological autopsy interviews are used for retrospective analysis in suicide research [25]. Various sources, such as statements by relatives and friends, doctors’ letters, and suicide notes, are used to analyze the motives for the suicide.

This study was approved by the Ethics Committee of Medical School Berlin (registration number: MSB-2021/67) on 14 July 2021.

### 2.2. Data Collection and Sample

Data were collected in July and August 2021 in Germany. Participants were mainly recruited via social media (e.g., Instagram, Twitter, Facebook) and self-help groups for the bereaved. The following criteria must have been met for participation: (1) lost a close male person (father, brother, partner, son) through suicide, (2) the suicide happened between three and twelve months ago, (3) 18 years or older, (4) sufficient German language skills, and (5) a signed consent form. A total of *N* = 15 participants were included.

### 2.3. Procedure

Interested participants first contacted the study team via email. Subsequently, these participants were sent the study information and the consent form. Once the consent form was signed and sent back, participants received a link to an online questionnaire which took about 15 min to complete. At the end of the questionnaire, participants could indicate preferred dates for the interview. Semi-structured interviews were conducted by telephone by a licensed clinical psychotherapist and were recorded using a recording device without an internet connection. At the beginning of the interview, the interviewer briefly introduced herself and explained the interview process. Participants could ask questions about the study and were explicitly informed that it was possible to take breaks if the interview was too stressful. On average, the interviews lasted 77.40 (*SD* = 23.38) minutes. As other types of information sources, such as doctors’ reports or suicide notes, are also analyzed during psychological autopsy interviews, relatives were asked to read them aloud during the interview. After the interview, participants were asked about how they were feeling and, if necessary, offered to talk further about their distress. If a participant had shown any problems with grief processing or general mental health, they would have been referred to further support services. The transcription of the audio files was based on the content-semantic transcription according to the method of Kuckartz [26] and was carried out with the MAXQDA Software 2022 [27]. All personal data were anonymized, and verbatim transcription was used. The language was slightly smoothed as the content, rather than the manner of speaking, was of relevance.

### 2.4. Quantitative Measures

The online questionnaire contained measures designed to assess the following domains: demographic information about the participant and the deceased, as well as symptoms of depression and anxiety of the deceased prior to suicide.

#### 2.4.1. Demographic Variables

Socio-demographic characteristics examined included birth year, gender, marital status, living situation, level of education, and relationship to the deceased. This information was collected for both the participants as well as for the deceased individuals. Additionally, information about the suicide and the mental health of the deceased prior to the suicide was assessed.

#### 2.4.2. Gender-Sensitive Depression Screening by Proxy (GDSD-25)

The GSDS-25 [28] is a self-assessment tool assessing depressive symptoms, especially those experienced by men. The questionnaire consists of 25 items, on the basis of which participants are asked to rate the intensity of depressive symptoms over the past weeks on a 4-point Likert scale ranging from *never or rarely* (0) to *mostly or always* (3). The questionnaire consists of six subscales: stress perception (five items), internal depressive symptoms (four items), aggressiveness (five items), emotional control (six items), risky behavior (two items), and alcohol abuse (three items). To calculate a total score, all values are added to a sum score. If the total score is greater than or equal to 19.5, depression is likely to be present. The GSDS-25 has proven good reliability with Cronbach’s alpha of α = 0.88 in a healthy student sample [29]. For the present study, a by-proxy version of the questionnaire was created so that participants answered the questions on behalf of the deceased. At the end, participants were asked to rate how well they could answer these questions about the deceased person. The by-proxy version showed good reliability with Cronbach’s alpha of α = 0.82.

#### 2.4.3. Short Version of the Patient Health Questionnaire by Proxy (PHQ-9)

The German version of the PHQ-9 [30] is a screening tool for assessing depressive symptoms, with 9 items rated on a 4-point scale (0 = *not at all* to 3 = *almost every day*), symptoms of anxiety and panic disorders with 5 items (yes/no answer format), and psychosocial impairment with 1 item (4-point rating scale from *not at all impaired* to *strongly impaired).* The items of the depression scale can be added up to a total score; a total score equal to or greater than ten indicates a major depression. The PHQ-9 showed good internal consistency with Cronbach’s alpha of α = 0.89 in a clinical sample [31]. A by-proxy version was also created here and showed good reliability with Cronbach’s alpha of α = 0.90 in the present sample.

### 2.5. The Psychological Autopsy Interview

The interview guide consisted of a total of six thematic categories and was based on existing autopsy studies and their implications [25,32,33]. The interview covered the following topics: (1) information about the suicide and the individual’s behavior prior to the suicide (e.g., “Was there some kind of announcement of the suicide or were there signs of the suicide?”); (2) mental health and therapy (e.g., “Did he experience therapy as helpful and what did he find helpful and unhelpful?”); (3) physical health (e.g., “Was he limited in engaging in hobbies/work or other leisure activities due to any physical illness?”); (4) social contacts (e.g., “Were there any conflicts with family or friends?”); (5) professional and financial situation (e.g., “Were there any work-related and/or financial problems?”); and (6) support for men at risk and gatekeepers (e.g., “What do you think would help men seek support when having suicidal thoughts?”).

### 2.6. Qualitative and Statistical Analyses

Quantitative data were analyzed using SPSS Version 27 while presenting frequencies for categorical variables, as well as means and standard deviations for continuous variables. A deductive–inductive approach was chosen for the evaluation of the qualitative data [34] considering the following steps: (1) preparing and organizing the respective dataset, (2) dividing the dataset into smaller samples, (3) creating codes while reading the first sample, (4) rereading the first sample while applying the codes, (5) analyzing the next sample while applying the codes, (6) creating new codes when the existing codes do not match, (7) recoding all samples once again while applying the new codes, (8) repeating the steps until the entire dataset is encoded. A category system consists of main categories and corresponding subcategories. The main categories are the deductive categories and are developed prior to analysis. The main categories were based on relevant literature and the existing interview guidelines. The interview guidelines consist of different thematic blocks (e.g., utilization and experience with support services), which form the main categories. Each main category usually contains several subcategories, which constitute the inductive category system. These are derived directly from the textual material following the mentioned steps. How many subcategories there are per main category varies and depends on the text content. The analysis was performed with the MAXQDA Software [27]. The material was independently coded by two people for reliability. Discrepancies were discussed with each other.

## 3. Results

### 3.1. Sample Characteristics

The total sample consisted of 15 female participants aged between 28 and 67 years with an average age of *M* = 42.73 (*SD* = 12.78) years. Six participants (40.0%) lost their brother, four (26.7%) their son, three (20.0%) their partner, and two (13.3%) their father. On average, the suicides happened 6.47 (*SD* = 2.39) months prior to the respective interview. Table 1 presents the sociodemographic data of the deceased.

### 3.2. Symptoms of Depression

On average, the reports regarding the deceased showed a total score of *M* = 24.07 (*SD* = 10.42) on the GSDS-25, which is above the suggested cut-off of 19.5 for depression. When looking at the individual scores of each person, 10 of the deceased (66.7%) scored above the proposed cut-off. Particularly high scores can be observed in the categories of emotional control (*M* = 9.47, *SD* = 4.90) and internal depressive symptoms (*M* = 5.47, *SD* = 3.14).

PHQ-D scores also indicate increased depressive symptoms overall, with *M* = 13.93 (*SD* = 7.29), with 10 of the deceased (66.7%) scoring above the suggested cut-off value of 10 for major depression.

### 3.3. Qualitative Findings

Four categories regarding the behavior prior to the suicide could be identified: (1) signs of suicide, (2) communication of suicidal thoughts and/or plans, (3) arrangements before suicide, and (4) mental health disorders.

#### 3.3.1. Signs of Suicide

This category includes any announcements of the suicide, or other signs of suicide, in the days leading up to the suicide. This category also covers whether the person changed significantly or behaved differently than usual. The results indicated that the majority of relatives (*n* = 13) noticed changes in the person but did not consider these changes to be warning signs for suicidal behavior. Often, men seemed more relaxed in the days before the suicide and were in a better mood than they had been previously. Relatives thus thought that the person was feeling better after a bad phase. In addition, men often tidied up, cleaned, or gave away their belongings in the days before the suicide.


*“He then started to clean everything. He also cleaned his car and the basement and so on. Normally, he would never have done that. At that moment, we thought, great, he’s feeling better now. Look, he even cleaned his car (…) I just didn’t know those could be signs for suicide.”*


Relatives also noted sudden changes regarding planning for the future. Men made comments that they would not grow old anyway or that their parents could not expect grandchildren from them. However, some men quit their jobs in which they had been employed for a long time without any problems or broke up with their partners and questioned their relationships.


*“He worked there for a long time, was always happy, had a good relationship with everyone. All of a sudden he went to work in the morning and quit. Just like that.”*


Other notable aspects were that affected men suddenly said goodbye to their relatives intensively in the evening, “*in case one of us doesn’t wake up anymore*”, or generally talked more about death. A large proportion of the deceased suffered from sleep problems in the days before the suicide. Only two of the relatives interviewed reported that they felt the changes might indicate suicidality.


*“I always said years ago that he would do something to himself one day. (...) It made sense to me.”*


#### 3.3.2. Communication of Suicidal Thoughts and/or Plans

This category includes whether the deceased men talked about suicide in general and whether they directly or indirectly addressed any suicidal thoughts or plans. Five relatives reported that the deceased person directly expressed suicidal thoughts. None of them disclosed any plans for suicide.


*“So he said again and again that he has suicidal thoughts. He said ‘I need to move out or otherwise I’m gonna kill myself’. I have to say, I never took it 100 percent seriously, of course.”*


Only three relatives reported indirect announcements of suicide.


*“So, he had made some comments that I had not taken seriously. He always said ‘If I have not achieved this and that by the age of 25, then I’ll kill myself.’ And that I would not have any grandchildren from him. We all have a very macabre sense of humor, so he just said, ‘Haha, you don’t really think that do you?’ And I really didn’t think that.”*


Indirect statements also included comments like “*I can’t do this anymore*” or asking relatives to “*let me go*”.

The other seven relatives reported that the deceased person never talked about suicide and never talked about dying or death in general. Some rather reported that their relative had previously expressed how one could even choose suicide as an option—indicating that they could not understand coming to such a decision.

#### 3.3.3. Arrangements before Suicide

This category includes arrangements that the deceased person made before the suicide; for example, managing their finances or preparing a will. In only five cases did the deceased actually make arrangements prior to their death. In this sample, this primarily included deleting emails and messages from dating profiles and passing on the passwords for the computer and all their accounts. In addition, some left letters to various people with instructions, had prepared or already canceled insurances, as well as transferred money to relatives.


*“My mother looked at her account and saw that my brother transferred all his money to her. He also had two accounts for Netflix and he canceled them. He also left exactly as much money on his bank account, as will be debited for bills. He left the exact sum on it.”*


All these preparations were noticed by the relatives only after the death. In some cases, the deceased also prepared their own funeral and wrote instructions to their relatives for this purpose. One participant reported that her son had suddenly given away belongings and that this had seemed “*strange*” to her.

Half of the participants reported that no preparations had been made by their deceased relative. Either the person had already been very organized before their death, or the person had apologized in his suicide note for not having arranged anything.


*“Normally (…) he was always super organized and has always arranged everything. But he didn’t even do that anymore and somehow apologized for it in his suicide note, saying I’m sorry for leaving such a mess behind.”*


#### 3.3.4. Mental Health

This category includes whether the deceased were diagnosed with a mental health disorder, which symptoms were present prior to the suicide, and whether there were any changes regarding symptoms. The quantitative data already showed that 13 of the deceased had been diagnosed with one or more mental health disorders: mainly depression, but also schizophrenia, and substance abuse. Relatives reported that mental health deteriorated significantly before the suicide, and progress made through therapy was no longer evident.


*“From the end of October, he had another very, very, very heavy phase of depression. And then he was definitely no longer the same person. So the two months he was still alive, he struggled a lot and suffered a lot and always said, we can’t imagine how bad it is.”*


In total, 12 of the deceased men primarily withdrew from family and friends, talked less, and complained of physical symptoms such as back pain, headaches, impaired vision, or difficulty swallowing. The majority (*n* = 8) reported sleeping disturbances and restlessness. This was also associated with increased anxiety due to physical symptoms. Three of the deceased men noticed changes in their bodies and could no longer distance themselves from catastrophizing thoughts. They were convinced that “*something is wrong, and they will die from the symptoms or have a bad disease*”.

In four cases, relatives reported psychotic symptoms in the days before the suicide. These included fears of being persecuted, fear of losing a job because of mistakes, and fear of harming someone. These fears took on such a dimension that the affected deceased were unable to distance themselves from them.


*“He wanted to turn himself in. He did a lot of things to us, he hid the keys for the store, he hid the car keys, he turned off the Internet (…) He took down the smoke detectors. He hid computers. In the end you were not allowed to be in the room with a cell phone because he was paranoid.”*


In six cases, relatives noted increased alcohol and/or drug use in the period preceding the suicide. Pre-suicide, the deceased drank significantly more alcohol, sometimes during the day and alone, which was normally unusual for them. When relatives approached them about consumption, it was usually trivialized. The drugs consumed included mainly cannabis, legal highs, and magic mushrooms.


*“He came into contact with cannabis. Then he took speed (...) then to be able to sleep, he started to buy benzodiazepines from someone. And of course everything with cannabis. Cannabis for the fun in between. And alcohol, too, of course. Always. Everything all together.”*


Some reported that the deceased individuals viewed drugs as a form of self-medication.


*“It gave him the idea that antidepressants don’t help at all and that mushrooms are the only thing that helps and that the pharmaceutical industry holds them against us and so on, and then he started to research all that and to question the normal antidepressants.”*


Aggressiveness and irritability are common symptoms of depression in men. Three men in this sample also showed these symptoms. Irritability was manifested primarily in affected individuals being “*annoyed and moody*”. In two cases, aggressive behavior occurred, resulting in physical fights and the involvement of the police.


*“He was highly aggressive. Then the police came. And (...) then they didn’t want to take him away. And I knew that if they left now, he would destroy everything here.”*


## 4. Discussion

This study aimed to assess changes in behavior and communication in men in the period preceding suicide, by interviewing the relatives bereaved by suicide, in the last 3 to 12 months. First, the quantitative analysis showed that two-thirds of the men who died showed signs of major depression in the weeks before the suicide. This finding is not surprising, as depression is associated with a significantly increased risk of suicide [35,36]. These findings were confirmed by the relatives’ information regarding the mental health disorders of the deceased in the qualitative interviews. A major problem, however, is that typical depression symptoms, such as crying, lethargy, and sadness, are generally known [37,38]. However, studies show that there is a certain gender bias here [15]. The symptoms mentioned are predominantly present in women in the context of depression. Since men often show externalizing symptoms when depressed, they are often not identified as at-risk individuals. This reinforces the need for gender-sensitive diagnosis and therapy.

Furthermore, this study evaluated if men showed any changes in the weeks before the suicide. In total, 13 relatives reported that the deceased behaved differently than usual in the period leading up to the suicide. The most common changes were an improved mood, being more relaxed, sometimes being more aggressive, and cleaning or tidying up. This finding is in line with other results of indicators of suicidal ideation and behavior, regardless of gender [39]. While the ambivalence between the wish to live and the wish to die can be extremely agonizing for those affected, the decision to attempt suicide is sometimes relieving, and the person appears more relaxed and in a better mood than in the days before. In addition, our study confirms the previous findings of Rasmussen et al. [40] and Rivlin et al. [41], who reported that men, in particular, showed an elevated mood, calmness, and relief in the days before the suicide. However, these are precisely the symptoms that make it especially difficult for relatives to recognize suicidal behavior. However, this phase is extremely dangerous, as it can indicate a decision to attempt suicide. This difficulty in identifying the suicidality of a close person was also confirmed by our results, as only two of the participants stated that they perceived these symptoms of calmness and positive mood in the prior days of suicide as warning signs. Only retrospectively did the remaining sample assess the symptoms as alarming. The findings are consistent with previous research regarding gatekeepers and knowledge about suicidality [23,42,43], as there often is a lack of knowledge of warning signs, especially regarding symptoms, in men.

Participants sometimes also reported increased alcohol and substance use, which was often unusual and surprising for the deceased person to that extent. As already confirmed in other studies, alcohol and/or drug abuse is a significant risk factor for suicidal experiences and behavior in men [2]. In addition, regular high alcohol consumption is closely associated with depressive symptoms, which can increase the risk of suicide [13,44].

Furthermore, we examined the extent to which deceased individuals talked about or announced suicide beforehand. One-third of respondents in our study reported that while men shared suicidal thoughts, none of the deceased revealed plans or direct intentions. Three of the relatives reported indirect announcements. Indirect statements often included comments such as “*I can’t take it anymore*” or “*it’s all getting too much for me*”, which are often not necessarily interpreted as suicidal ideation. About half reported that there was no announcement. These findings contradict the widespread assumption that most men do not communicate at all their suicidal thoughts. While men communicate their thoughts about suicide, they do so in a coded and indirect way so that most relatives are unable to recognize them as holding serious intent [19]. These findings are consistent with other studies on communication of distress and suicidal ideation [19,38]. Men are more reluctant than women to disclose suicidal crises. However, this does not only refer to the immediate social environment. In contact with the health care system, men also disclose suicidal thoughts less often and openly than women [45]. Men show more restraint here, mainly due to shame.

Managing finances, making a will, or other preparations may also indicate suicidal ideation. In our sample, only five relatives reported that the men transferred money or shared passwords beforehand. Other studies have also shown that only a small proportion of men make preparations before attempting suicide. Sinyor et al. [46] examined 285 suicide notes and only one-fifth consisted of will content. Similarly, in the study by Rivlin et al. [41], only 12% of men made preparations or gave instructions in their suicide notes. Thus, it appears that while preparations may be a non-negligible sign, men rarely tend to discuss last wills or financial matters with loved ones, and these arrangements are not discovered until after a suicide.

Lastly, the manifestation of a mental health disorder or change in symptoms of an existing mental health disorder may also imply suicidal experience and behavior. In our sample, the majority of the deceased (86.7%) had a diagnosis of mental health disorder, some of which existed in previous years, but others manifested shortly before the suicide. Here, the majority of relatives reported that symptoms worsened or new symptoms, some of them psychotic, emerged in the weeks before the suicide. Frequently, the affected men also withdrew from their social environment and showed anger or aggression. These findings are also in line with other studies on male suicidality [47,48,49]. Player et al. [49] interviewed 35 male suicide survivors as well as 47 relatives of male suicide survivors and found that almost all men showed symptoms of anger, aggression as well as violence, and social isolation. The results also confirm that men show more externalizing symptoms in the context of depression and that these are less recognized as such [7,15], which highlights the need to pay more attention to this symptomatology in order to prevent suicides in men in the long term.

This study has both strengths and limitations. Among the strengths, the time criterion of 3 to 12 months ensured that relatives could more easily recall the time before the suicide. Autopsy interviews are also a common method for reconstructing the history of a suicide as accurately as possible. In addition, our sample has a sufficient variance regarding the degree of kinship and the age of the deceased.

A limitation of this study is the relatively small sample of 15 individuals, which makes it difficult to generalize our findings, specifically the quantitative data analyses which have been included. Impressions and statements may be biased by the interviews, as they were collected retrospectively. Recall bias is a well-known phenomenon in autopsy interviews [50]. Moreover, only one person was interviewed per case and not the whole family system. Thus, when interpreting the results, the aspect of the subjective perception of this one person should be taken into account. In addition, it must be considered that all relatives in this study were highly involved in the life of the deceased person, supporting and helping them throughout the crisis. This population group is often strongly motivated to participate in studies as they want to make a difference. However, it would still be interesting to record the experiences of less involved relatives or of different people in the social environment.

## 5. Conclusions

Men show changes in their behavior prior to suicide. While these changes are noticed by relatives, they are often not identified as alarming. Moreover, men do seem to communicate suicidal intentions, but unfortunately, mostly in an indirect manner. Therefore, these statements are not recognized as suicidal intentions by their environment. Since men are much less likely to turn to professional support systems, it is essential that surrounding individuals have a comprehensive knowledge of the suicidal experiences and behavior of men. Through this, men at risk could be more quickly identified and connected to the relevant support systems. Psycho-educational gatekeeper programs seem to be an important component in suicide prevention and can help to sensitize potential contacts of men at risk. Furthermore, increased media reports about these specific communication patterns and behavior changes in men prior to suicide could improve knowledge across the general population.

## Figures and Tables

**Table 1 ijerph-20-06668-t001:** Sociodemographic data of the deceased men (*N* = 15).

	*M (SD)*	Range
Age in years	38.47 (14.85)	17–75
	*n*	%
Gender (male)	15	100
Marital status		
Single	8	53.3
In a relationship	2	13.3
Married	4	26.7
Divorced	1	6.7
Living situation		
Alone	5	33.3
With parents	6	40.0
With partner	4	26.6
Method of suicide		
Rail suicide	1	6.7
Hanging	6	40.0
Intoxication	1	6.7
Jumping from a height	4	26.7
Other (drowned, suffocated)	3	20.0
Farewell letter (yes)	4	26.7
Prior suicide attempts (yes)	4	26.7
Diagnosis ^1^	13	86.7
Depression	10	66.7
Schizophrenia	2	13.3
Substance abuse	3	20.0
Anxiety disorder	1	6.7
In therapy at time of death	5	33.3
Therapy in the past	10	66.7
Psychotropic medication	7	46.7

Note: ^1^ multiple answers possible.

## Data Availability

The data can be requested from the first author.

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
