# Peer review of "Men’s Behavior and Communication in the Days Prior to a Suicide—A Psychological Autopsy Study"

_ijerph, 2023, doi:10.3390/ijerph20176668_

Round 1

Reviewer 1 Report

It is a manuscript that addresses a prominent issue of suicidal behavior in men.

The abstract does not clearly indicate the aim of the study, nor the methodology used.

The introduction is suitable for the study since it accurately points out the lack of knowledge about the object of study.

The methodology is adequately described as related to the quantitative aspect. However, the description of the qualitative analysis is unclear. It is necessary to give more detail on the methodology of the qualitative analysis. For example, How the identified themes were generated? How many coders participated in the process? What procedure was used to guarantee the internal reliability of the codings? It is mentioned that a deductive-inductive approach was used, but it is unclear how each was done. What deductive aspects were incorporated into the study? An important limitation of the qualitative study was that only one relative was interviewed for each case. It is necessary to include it in the limitations.

In the results section, the qualitative part is clear. However, the qualitative section requires substantial improvement. It is worrisome to observe in the writing of the findings that the relatives were incapable of identifying alarm signals when the study aims to evaluate the changes in the behavior and communication of the man before the suicide. The wording of the findings is more focused on relatives than on meeting the aims. The information on the topics is redundant in certain respects. It is necessary to consider in which section of the findings each type of information should be included. A recurring comment was that the relatives were unable to identify the alarm signals. A description more focused on men and less on relatives is necessary.

The discussion, sadly, goes to arguments already used regarding the warning signs before suicide in other studies. The authors insist on the argument of the discussion to the lack of capacity of the relatives to identify the alarm signals. Furthermore, they fail to delve into a critical discussion of men's behaviors and communications (perhaps using a gender perspective). There is no clarity in the contributions of the study compared to other investigations on the subject.

Reviewer 2 Report

Thank you for the opportunity to review your article. Your presentation of your study was very well done. I appreciated your inclusion of direct quotes from the family members, as I think this is valuable for anyone reading your research article.

I did not find much to change, as I think you all did a fantastic job on this article. I would suggest adding an explanation of what a "gatekeeper" is (who they are). This would allow anyone reading your article to understand that terminology being used in case they aren't familiar with that terminology.

Reviewer 3 Report

I think this is a good study on the signs of suicide in men and behavioral changes before suicide. 

It would be better to provide a little more detailed explanation of the analysis method.

In “2.6. Qualitative and Statistical Analyses”, main categories and subcategories were distinguished, and it would be nice if authors explained a little more about the contents corresponding to these categories.

None.

Reviewer 4 Report

In this interesting manuscript entitled Men’s Behavior and Communication in the Days prior to a Suicide – A Psychological Autopsy Study, authors investigated behavior prior to the suicide of 15 suicide males victims using a mixed-methods approach. Qualitative data were collected conducting psychological autopsy interviews with 15 females who lost a male relative or partner to suicide. Quantitative data were obtained prior to the interviews using online questionnaires. Signs of suicide, communication of suicidal thoughts and/or plans, arrangements before suicide and mental health disorders of suicide victims were assessed.

Main contributions and strengths:

-          Importance of the manuscript as this  phenomenom is not yet fully researched, especially among males, and is usually unrecognised and underestimated;

-       Possible contribution to preventive activities in men with risk for suicide;

-          The time criterion (3 - 12 months after suicide) still ensure recall the time before the suicide in the relatives;

-          Autopsy interviews are wellknown best practice postmortem procedure or tool used in suicide, especially in suicide cases in which there were no prior indications or signs of suicidal thoughts or behaviors;

-          Sample variance (e.g. degree of kinship, age of the deceased);

Limitations:

-        Relatively small sample of 15 males suicide victims and 15 females relatives of suicide victims - difficult to generalize findings to the whole general population;

-       Retrospective collection of subjective information, data and impressions: problem of objectivity;

-       Retrospective collection of subjective information, data from relatives (not suicide subject) (misunderstanding, unrecognition);

-          Autopstic interviews: problem of recall bias;

The manuscript is clear, relevant for the field and presented in a structured manner and scientifically sound. The manuscript’s results are reproducible based on the details given in the methods section.

The introduction provides sufficient background, however I suggest authors to add information and references for readers: that women are more likely than men to attempt suicide and to have suicidal thoughts, men are more likely than women to die by suicide, men are less likely to share their emotional distress and they are often reluctant to seek help, men more likely self treat symptoms of depression with alcohol and other substances.  

The research design is appropriate.

Materials and methods are adequately described.

Regarding Ethics Review: authors reported that the study was conducted in accordance with the Declaration of Helsinki, and approved by the Ethics Committee of the Medical School Berlin (registration number: MSB-2021/67) on July 14th, 2021. Informed Consent Statement was obtained from all subjects involved in the study.

Under 2. Materials and Methods 83 - 2.3. Procedure 104 authors described that …After the interview, participants were asked about how 117 they were feeling and, if necessary, offered to talk further about their distress… However I would like to ask authors, in case if there were a need of further clinical asssesment if the subject was informed about or adressed to more specific and further treatment? Please add.

Results are clearly presented.

The table is appropriate and properly show the data, the data are interpreted appropriately and consistently throughout the manuscript.

The conclusions are supported by the results, and are consistent with the evidence and arguments presented.

Regarding the References I would like to suggest to update bibliography as among 48 references less than a half are from last 5 years. For example in 4. Discussion is appropriate to add that depression and substance use are most frequently in men suicide victims. I would like to suggest to add sentence explaining that calmer mood in men with suicide risk can be also a cause of alarm of suicide risk (made decision on suicide and is at peace with it).

Round 2

Reviewer 1 Report

Thanks to the authors for the corrections of the manuscript.

The most significant objections I had regarding the manuscript have been remedied.